# The Association between Acute and Late Genitourinary and Gastrointestinal Toxicities: An Analysis of the PACE B Study

**DOI:** 10.3390/cancers15041288

**Published:** 2023-02-17

**Authors:** Ragu Ratnakumaran, Victoria Hinder, Douglas Brand, John Staffurth, Emma Hall, Nicholas van As, Alison Tree

**Affiliations:** 1The Royal Marsden NHS Foundation Trust, London SM2 5PT, UK; 2Radiotherapy and Imaging Division, Institute of Cancer Research, London SM2 5NG, UK; 3The Institute of Cancer Research, Clinical Trials and Statistics Unit, London SM2 5NG, UK; 4Department of Medical Physics and Bioengineering, University College London, London WC1E 6BT, UK; 5Division of Cancer and Genetics, School of Medicine, Cardiff University, Cardiff CF14 4XN, UK

**Keywords:** acute toxicity, late toxicity, prostate cancer, stereotactic body radiotherapy, intensity-modulated radiotherapy, gastrointestinal, genitourinary

## Abstract

**Simple Summary:**

Several studies have shown the association between significant short-term and long-term side-effects after prostate radiotherapy using older techniques. Our study, tests whether there is an association between short- and long-term bowel and urinary side-effects with modern prostate radiotherapy techniques such as stereotactic body radiotherapy (SBRT) and intensity modulated radiotherapy (IMRT). We use CTCAE clinical assessments of patient symptoms for radiotherapy side-effects in the PACE-B study to answer this question. We show that patients who develop short-term urinary and bowel side-effects are at higher odds of developing long-term side-effects, after conventional fractionated radiotherapy and SBRT. This association remains even after adjusting for patient, treatment and tumour factors. We show that patients who have significant urinary symptoms before radiotherapy are also at higher odds of developing long-term side-effects. We suggest that patients who experience significant short-term side-effects should be closely monitored and potentially have their symptoms treated earlier.

**Abstract:**

Several studies have demonstrated the association between acute and late radiotherapy toxicity in prostate cancer using older radiotherapy techniques. However, whether this association is present with newer techniques such as stereotactic body radiotherapy (SBRT), remains unclear. We use univariable and multivariable logistic regression to analyse the association between grade 2 or worse acute gastrointestinal (GI) and genitourinary (GU) toxicities with equivalent late toxicities in patients treated with SBRT and conventional or moderately fractionated radiotherapy (CRT) within the PACE-B study. 842 patients were included in this analysis. Common Terminology Criteria for Adverse Events (CTCAE) was the primary clinician reported outcome measure used in this analysis. In univariable analysis, experiencing a grade 2+ acute GU toxicity was significantly associated with developing a grade 2+ late GU toxicity after SBRT (OR 4.63, 95% CI (2.96–7.25), *p* < 0.0001) and CRT (OR 2.83, 95% CI (1.69–4.71), *p* < 0.0001). This association remained significant in multivariable analysis. In univariable analysis, experiencing a grade 2+ acute GI toxicity was also associated with developing a grade 2+ late GI toxicity after SBRT (OR 3.67, 95% CI (1.91–7.03), *p* < 0.0001) and CRT (OR 4.4, 95% CI (2.04–9.47), *p* < 0.0001). This association also remained significant in multivariable analysis. Grade 2+ baseline GU symptoms were also associated with grade 2+ late urinary toxicity in both univariable and multivariable analysis. Overall, acute toxicity is an important predictor variable for late GU/GI toxicity after localised prostate radiotherapy using SBRT and CRT. Future work should test whether optimising symptoms pre-treatment and early intervention in those with significant acute toxicities could mitigate the development late of toxicity.

## 1. Introduction

Prostate cancer is one of the most frequently diagnosed non-cutaneous malignancies affecting men globally, with nearly 1.5 million new cases each year [1]. Many men present with National Comprehensive Cancer Network (NCCN) low or intermediate-risk prostate cancer, which is highly curable [2]. A range of proven treatments are available, where treatment is required, including surgery, brachytherapy, external beam radiotherapy. For early prostate cancer, all treatment options have similar tumour control rates at ten years but differing effects on sexual dysfunction, gastrointestinal (GI) and genitourinary (GU) side effects [3,4]. Radiotherapy approaches for localised prostate cancer have transformed over the last 20 years. Due to technological advances and improvement in image guidance, radiotherapy techniques have evolved from traditional 3D conformal approaches to intensity-modulated radiotherapy (IMRT). As techniques have improved, so has our knowledge of the radiobiology of prostate cancer, and it is now recognised that prostate cancer has a low alpha-beta ratio (<2 Gy) [5,6]. This has allowed us to increase fraction size to take advantage of the fraction size sensitivity of prostate cancer. Trials such as CHHiP and PROFIT tested moderately fractionated radiotherapy (3 Gy per fraction) against the traditional 2 Gy per fraction and demonstrated equivalent biochemical free failure rate and toxicity outcomes [7,8]. These practice-changing trials have established 60 Gy in 20 fractions as the standard recommended regime for localised prostate cancer in the UK. However, we are now in the era of ultra-hypofractionated radiotherapy, delivering higher doses per fraction with improved image guidance. The HYPO-RT-PC trial tested 42.7 Gy in 7 Fr compared with conventional radiotherapy and demonstrated non-inferiority with ultra-fractionated radiotherapy in terms of biochemical-free survival and toxicity [9]. Though most patients were treated with conformal techniques, it demonstrates that ultra-fractionated schedules are feasible and safe.

PACE-B is the first multi-centre international phase 3 randomised control trial, comparing ultra-hypofractionated radiotherapy using stereotactic body radiotherapy (SBRT) against conventional or moderately hypofractionated radiotherapy (CRT). The study has reported no difference in Radiation Therapy Oncology Group (RTOG) GI and GU toxicity at 12 weeks between the treatment arms. Similarly, there is no difference in GI and GU toxicity measured using RTOG clinician-reported outcome at 24 months [10,11]. However, higher GU toxicities were noted after SBRT when using the Common Terminology Criteria for Adverse Events (CTCAE) clinician-reported outcomes, which was also concordant with data from patient-reported outcomes [11]. Several studies have suggested an association between acute and late toxicity with conventionally fractionated radiotherapy [12,13,14,15,16,17]. Those who experience significant acute toxicity events were shown to be at a higher risk of developing late side effects. It has been postulated that this could either be due to shared propensity for developing acute and late toxicity due patient factors (e.g., genetic), significant baseline symptoms, dosimetric risk factors and consequential late effects [12,13,18]. In the CHHiP trial, higher acute GI toxicity was noted in the 60 Gy in 20 Fr arm, however this did not translate to higher late GI toxicity, therefore further exploration of this relationship with new fractionation schedules is warranted. Recognising those at risk of significant late side effects earlier could provide an opportunity for early intervention; however, whether this association is present after ultra-hypofractionated radiotherapy is currently unknown.

This paper assesses the association between acute GI/GU toxicity and late GI/GU toxicity after SBRT and CRT, while considering patient, tumour and treatment factors, in patients treated within the PACE-B study. It should be noted that GI/GU toxicities are based on patient clinical assessments of symptoms using CTCAE criteria, which do not need to be attributed to radiotherapy treatment. Given the shift towards using patient reported outcome measures as the primary toxicity endpoint, we also look at the association between CTCAE and patient reported outcome measures.

## 2. Materials and Methods

### 2.1. Study Design and Participants

The PACE-B trial is a phase III non-blinded multicentre non-inferiority randomised controlled trial. The trial ran in 35 centres in three countries (UK, Ireland and Canada) and aims to assess non-inferiority of SBRT compared with CRT. The primary endpoint is freedom from biochemical and clinical failure at 5 years, and secondary endpoints include clinician-reported and patient-reported outcome measures in the acute and late time-period. The PACE-B study only recruited patients who had chosen radical radiotherapy as their preferred treatment modality. The study population and eligibility criteria are summarised extensively elsewhere, but in short, patients with WHO performance status 0–2 and histological confirmed prostate adenocarcinoma with low-risk or intermediate risk disease were included [10,11]. Participants in the PACE-B study did not receive androgen deprivation therapy. Here we analyse patients with at least one CTCAE measurement for GI and GU events in both the acute and late time period. The acute period is defined as the assessments from end of treatment to 12 weeks post-radiotherapy and the late period from the assessments 6–24 months post radiotherapy. Data beyond 24 months is not currently available. Visit data was used based on the visit assigned to by sites and not based on the actual date of the visit.

### 2.2. Radiotherapy Details

The details of radiotherapy techniques have been previously published [10,11]. In summary, fiducial insertion was recommended for all participants at least 7 days before their CT planning scan. A radiotherapy planning MRI was also strongly recommended, but not mandatory. The clinical target volume (CTV) for low-risk disease patients included the prostate only, while for intermediate-risk disease the prostate and proximal 1 cm of seminal vesicle were included. For CRT, the recommended CTV to planning target volume (PTV) expansion was a 5–9 mm isometric, except posteriorly where it was 3–7 mm. For SBRT, the CTV to PTV expansion was 4–5 mm isometric, expect 3–5 mm posteriorly.

Patients in the CRT arm received 78 Gy in 39 Fractions over 7–8 weeks or following an approval protocol amendment (on 24 March 2016), 62 Gy in 20 fractions over 4 weeks. This change followed publication of the CHHiP trial data which supported moderate hypofractionation, but a higher dose (62 Gy vs. 60 Gy) was chosen as PACE-B prohibits androgen deprivation therapy. For the SBRT arm, the PTV dose was 36.25 Gy in 5 fractions and 40 Gy to the CTV over 1 to 2 weeks. Image guided radiotherapy was mandated (either fiducials or cone beam imaging). For SBRT, intrafraction motion monitoring was permitted, otherwise repeat static imaging was necessary for treatment delivery beyond 3 min.

### 2.3. Follow-Up

Clinician-reported outcomes (CROs) were measured at baseline, at alternative weeks during CRT and at the final fraction for SBRT. CROs were assessed after treatment at weeks 2, 4, 8 and 12, and then three monthly for the first 2 years and six monthly until year 5.

Clinicians completed both RTOG (GI and GU domain) at baseline and every visit, and the CTCAE (Version 4.03) at baseline, end of treatment (for SBRT arm), and the remaining subsequent follow-ups. CTCAE items were graded from 0 to 5.

Patient-reported outcome measures were collected, including the Expanded Prostate Cancer Index Composite Short Form (EPIC-26) and International Prostate Symptom Score (IPSS). EPIC-26 scores were assessed at baseline, weeks 4 and 12, and at months 6, 9, 12 and 24 months, and annually thereafter. IPSS score was measured at baseline week 2, 4, 8 and 12 and at months 6, 9, 12 and 24 months and annually thereafter.

### 2.4. Statistical Analysis

For this analysis, we primarily analyse CTCAE outcome measures due to its increased sensitivity in the acute and late assessment in PACE-B [10,11]. Specific CTCAE items for GI symptoms include colitis, constipation, diarrhoea, GI fistula, nausea, proctitis, haemorrhage and rectal pain. For GU symptoms, specific items include haematuria, pain/dysuria (cystitis), frequency, incontinence, urinary retention and urgency. EPIC-26 scores provide a quality-of-life measure for GI, GU, sexual and general domain. EPIC-26 were re-scaled to a 0 to 100-point scale, with a higher score associated with better quality of life scores [19,20]. The IPSS is made up of 7 questions related to urinary symptoms. A score of 0 indicates no symptoms, 1–7 points indicates mild, 8–19 points for moderate, and 20–35 points for severe urinary symptoms. In this analysis we look at IPSS total score, EPIC-26 overall urinary bother score, and EPIC-26 bowel sub-domain score. This analysis focused on GU and GI toxicities, and sexual dysfunction was not assessed.

Logistic regression univariable and multivariable models were fitted to assess factors associated with late CTCAE events GU and GI events. Two definitions for late toxicity were used; any late GI or GU grade 2+ during the late toxicity period and persistent late toxicity defined as grade 2+ toxicity recorded at two or more timepoints during the late period. A sensitivity analysis was also performed using late toxicity (12–24 months) to see if the results were concordant when any potential residual acute toxicity was excluded. Univariable and multivariable logistic regression models were used to assess the association between patient factors, tumour factors, treatment factors and baseline and acute toxicity on CTCAE grade 2+ late toxicity. Factors assessed were: patient age at randomisation, NCCN risk group, prostate volume, fiducial use, alpha-blocker/anti-cholinergic use before randomisation, SBRT treatment modality (CyberKnife vs. conventional linac for SBRT patients only). The following toxicity measures were assessed; acute baseline toxicity (worst grade 2+ event), baseline and acute IPSS score, baseline and acute EPIC-26 urinary bother, and EPIC-26 bowel sub-domain score, and persistent acute toxicity (2 or more grade 2+ events). Logistic regression models were performed separately for SBRT and CRT patients. Number of late events by possible associated factors are presented for the univariable analysis and the odds ratio (and 95% confidence interval) for each variable and their corresponding *p*-value are reported. Lasso variable selection method was used to determine variables to include in the multivariable logistic model [21]. Predictor variables with missing data (greater than 10% missing) or potential collinearity were excluded from lasso variable selection. The area under receiver operator characteristics (ROC) curve (AUC) was used to evaluate the predictive performance of logistic regression models. True positive and false positive rates are reported to indicate the accuracy of acute grade 2+ toxicities in predicting late grade 2+ toxicities.

Agreement between worst grade 2+ CTCAE events and worst IPSS and EPIC-26 bowel sub-domain score and urinary bother scores were also assessed in the acute and late time period using weighted Cohen’s kappa statistics. The scores were converted into three categories. The first category represents no or minimal toxicity (CTCAE score of 0; IPSS score 0–7: EPIC-26 bowel/urinary bother >75), the second category represents mild/moderate toxicity (CTCAE score of 1; IPSS score 8–19; EPIC 26 bowel/urinary bother >25 and ≤75), and the third category represents severe toxicity (CTCAE score or 2 or more; IPSS score 20–35; EPIC-26 bowel/urinary bother ≤25). Using the cut-off suggested by Landis and Kock, kappa over 0.81–1.00 reflects perfect agreement, 0.61–80 reflects substantial agreement, 0.41–0.60 reflects moderate agreement, 0.21–0.40 reflects fair agreement, 0.00–0.20 reflect slight agreement and less than 0 reflects poor agreement [22]. Boxplots were also produced to demonstrate the relationship between patient reported outcome measures and GI/GU CTCAE. The data was analysed based on available data and does not account for missing data. Summary statistics on the completeness of data are presented. The percentage of CTCAE assessment visits that fall within the schedule visit window based on the visit window being half-way between consecutive visits will be presented. In addition, the percentage of CTCAE assessment visits that fall within the correct acute and late time periods will be presented. All statistical analyses were performed using STATA (version 17). The PACE study is prospectively registered at ClinicalTrials.gov (accessed on 17 December 2022), NCT01584258.

## 3. Results

842 patients had at least one CTCAE outcome measurement recorded in the acute and late periods. 414 patients were treated with SBRT, while 428 were treated with CRT. The median follow-up time from end of treatment was 24.1 months (interquartile range [IQR] 23.6–24.7). Table 1 summarises the pre-treatment and baseline characteristics. Although baseline CTCAE grading was intended to record grade 2 for those on alpha blockers and anticholinergics, the number of patients recorded to being on these medications (19–20% and 4%, respectively) is larger than the number recorded as having grade 2+ CTCAE symptoms at baseline (5% for SBRT in Table 2 and 4% in CRT in Table 3). This discrepancy was noticed by one of the reviewers. Completeness of CTCAE assessments and patient reported outcomes by visit is presented in the Appendix A (Appendix A). Regarding the completeness of data within a patient level, in the acute time period, CTCAE data was collected for at least three visits (the maximum was four visits for the CRT group and five visits for the SBRT group) for 97% of all patients. In the late period, data was collected for at least five visits (the maximum was seven visits) for 92% of all patients (Appendix A). Ninety-five percent (8361/8844) of the assessment visits fell within the scheduled visit window. Less than 1% of CTCAE assessments fell outside of the relevant acute and late time periods.

### 3.1. Association between Acute and Late Genitourinary Toxicity

#### 3.1.1. SBRT

One hundred and thirty-two patients (32%) reported at least one grade 2+ late GU toxicity (6–24 months) after SBRT. Fifty-five percent (true positive rate) of these patients reported at least one grade 2+ acute GU toxicity. Of the 282 patients (68%) without any grade 2+ late GU toxicity, 21% (false positive rate) reported at least one grade 2+ acute GU toxicity. Seventy-three patients (18%) reported persistent grade 2+ late GU toxicity after SBRT. Sixty-five percent (true positive rate) of these patients experienced at least one grade 2+ acute GU toxicity. Of the 341 patients without persistent grade 2+ late GU toxicity, 24% (false positive rate) experienced at least one grade 2+ acute GU toxicity (Figure 1).

Univariable and multivariable logistic regression for predicting a grade 2+ late GU toxicity (6–24 months) and persistent grade 2+ GU toxicity in the SBRT population is summarised in Table 2. In a univariable analysis baseline grade 2+ GU events (*p* < 0.0001), baseline IPSS scores (*p* < 0.0001) and baseline EPIC-26 overall urinary bother (*p* < 0.0001) were statistically significantly associated with late grade 2+ GU toxicity after 6 months. Worst acute grade 2+ GU toxicity, persistent grade 2+ acute GU toxicity, worst acute IPSS total score, and worst EPIC-26 acute overall urinary bother score were all strongly associated with experiencing a late grade 2+ GU toxicity (*p* <0.0001). Univariable analysis also demonstrated that acute grade 2+ urinary obstructive symptoms (*p* < 0.0001), irritative symptoms (*p* < 0.0001), pain/dysuria (*p* < 0.0001), and incontinence (*p* = 0.001) were individually associated with late urinary toxicity. The remaining baseline and treatment factors showed no statistical association, except for prostate volume (*p* = 0.02) and the use of conventional linac (CL) (*p* = 0.002). In multivariable logistic regression, baseline grade 2+ GU toxicity (OR 4.27, CI (1.41–13.04, *p* = 0.01)), worst acute grade 2+ GU toxicity (OR 3.70, CI (2.39–5.98), *p* <0.0001), SBRT-CL (OR 2.30, CI (1.34–3.94), *p* = 0.002) and fiducial use (OR 2.44 CI (1.35–4.40), *p* = 0.002), were statistically associated with grade 2+ late GU (6–24 months) toxicity. The area under the receiver operating characteristic (ROC) curve (AUC) for the multivariable model’s classification performance for grade 2+ late GU (6–24 months) toxicity was 0.73 (95% CI, 0.67–0.78).

In multivariable analysis baseline grade 2+ GU toxicity, worst acute grade 2+ GU toxicity, and fiducial use remained predictive of persistent grade 2+ late urinary toxicity. The AUC for the multivariable model to predict persistent grade 2+ late urinary toxicity was 0.77 (95% CI, 0.70–0.83). Univariable and multivariable logistic regression for predicting worst grade 2+ late GU toxicity (12–24 months) is summarised in Appendix A in the Appendix A.

#### 3.1.2. CRT

Eighty-three patients (19%) reported at least one grade 2+ late GU toxicity (6–24 months) following CRT. Forty-one percent (true positive rate) of these patients reported at least one grade 2+ acute GU toxicity. Of the 345 patients (81%) without any grade 2+ late GU toxicity, 20% (false positive rate) reported at least one grade 2+ acute GU toxicity. Twenty-six patients (6%) reported persistent grade 2+ late GU toxicity after CRT. Sixty-six percent (true positive rate) of these patients experienced at least one grade 2+ acute GU toxicity. Of the 404 patients (94%) without persistent grade 2+ late GU toxicity, 21% (false positive rate) experienced at least one grade 2+ acute GU toxicity (Figure 2).

Univariable and multivariable logistic regression for predicting worst grade 2+ late GU toxicity (6–24 months) and persistent grade 2+ late GU toxicity in the CRT population is presented in Table 3. In the univariable analysis, baseline grade 2+ GU events (*p* < 0.0001), baseline IPSS total score (*p* = 0.009) and baseline EPIC-26 overall urinary bother score (*p* < 0.0001) are associated with grade 2+ late GU toxicity (6–24 months) in the CRT population. Worst acute grade 2+ GU events (*p* < 0.0001), persistent acute grade 2+ GU (*p* < 0.0001), worst IPSS total score (*p* < 0.0001) and worst EPIC-26 urinary bother score (*p* < 0.0001) were all individually associated with experiencing grade 2+ late GU toxicity. Univariable analysis also demonstrated that acute grade 2+ urinary obstructive symptoms (*p* < 0.0001), irritative symptoms (*p* = 0.003) and incontinence (*p* = 0.01) were all separately associated with late urinary toxicity. The remaining baseline and treatment factors showed no statistical association, except for prostate volume (OR 1.02, CI (1.01–1.03), *p* < 0.002). In multivariable logistic regression, baseline grade 2+ GU toxicity (OR 6.73, CI (2.19–20.7), *p* = 0.001), grade 2+ acute GU toxicity (OR 2.26, CI (1.30–3.95), *p* = 0.004), and prostate volume (OR 1.02, CI (1.00–1.03), *p* = 0.009) remained statistically significantly associated with grade 2+ late GU toxicity (6–24 months). The AUC for the multivariable model’s ability to predict grade 2+ late GU (6–24 months) toxicity after CRT was 0.66 (95% CI, 0.59–0.73).

In multivariable analysis to predict persistent late grade 2+ GU events after CRT, baseline grade 2+ GU toxicity (*p* = 0.001) and worst acute grade 2+ GU toxicity (*p* = 0.001) remained statistically significant. In univariable analysis, fiducial use was protective for late GU toxicity following CRT (OR 0.30, CI (0.13–0.69), *p* = 0.005), however it was not significant in multivariable analysis. The AUC for the multivariable model’s ability to predict grade persistent 2+ late GU toxicity after CRT was 0.77 (95% CI, 0.67–0.87). Univariable and multivariable regression for predicting grade 2+ GU late toxicity after 12 months following CRT is summarised in Appendix A of the Appendix A.

### 3.2. Association between Acute and Late Gastrointestinal Symptoms

#### 3.2.1. SBRT

Fifty-one patients (12%) reported at least one grade 2+ late GI toxicity (6–24 months) after SBRT. Thirty-five percent (true positive rate) of these patients reported at least one grade 2+ acute GI toxicity. Of the 363 patients (88%) without any grade 2+ late GI toxicity, 13% (false positive rate) reported at least one grade 2+ acute GI toxicity. Twenty-two patients (5%) reported persistent grade 2+ late GI toxicity after SBRT. Forty-one percent (true positive rate) of these patients experienced at least one grade 2+ acute GI toxicity. Of the 392 patients (95%) without persistent grade 2+ late GI toxicity, 15% (false positive rate) experienced at least one grade 2+ acute GI toxicity (Figure 3).

Univariable and multivariable logistic regression for predicting worst grade 2+ late GI toxicity (6–24 months) and persistent grade 2+ late GI toxicity in the SBRT group is presented in Table 4. In the univariable analysis baseline EPIC-26 bowel sub-domain score was predictive for late grade 2+ GI toxicity (*p* = 0.004). Acute grade 2+ GI events (*p* < 0.0001) and worst EPIC-26 bowel sub-domain score in the acute setting (*p* = 0.04) was also predictive for grade 2+ late GI toxicity. In univariable analysis, acute grade 1+ rectal bleeding (*p* = 0.04), grade 2+ diarrhoea (*p* = 0.01) and grade 1+ rectal pain (*p* < 0.0001) were individually associated with late rectal toxicity. The remainder of the baseline and treatment factors did not demonstrate a statistically significant association. In multivariable analysis, acute grade 2+ GI toxicity (OR 3.68, CI (1.89–7.17), *p* < 0.0001) remained statistically significantly associated with grade 2+ late GI toxicity (6–24 months). The AUC for multivariable models’ ability to predict grade 2+ late GI (6–24 months) toxicity after SBRT was 0.66 (95% CI, 0.57–0.75). Univariable analysis for predicting persistent late grade 2+ late GI events is summarised in Table 4. Due to low number of events multivariable regression was not performed. Appendix A in the Appendix A summarises logistic regression analysis to predict late grade 2+ GI toxicity after 12 months.

#### 3.2.2. CRT

Fifty-two patients (12%) reported at least one grade 2+ late GI toxicity (6–24 months) after CRT. Twenty-three percent (true positive rate) of these patients reported at least one grade 2+ acute GI toxicity. Of the 376 patients (88%) without any grade 2+ late GI toxicity, 7% (false positive rate) reported at least one grade 2+ acute GI toxicity. Eighteen patients (4%) reported persistent grade 2+ late GI toxicity after CRT. Seventeen percent (true positive rate) of these patients experienced at least one grade 2+ acute GI toxicity. Of the 412 patients (96%) without persistent grade 2+ late GI toxicity, 8% (false positive rate) experienced at least one grade 2+ acute GI toxicity (Figure 4).

Univariable and multivariable logistic regression for predicting worst grade 2+ late GI toxicity (6–24 months) and persistent grade 2+ late GI toxicity in the CRT population is presented in Table 5. In univariable analysis, experiencing an acute grade 2+ GI events (*p* < 0.0001), persistent acute grade 2+ GI toxicity (*p* = 0.003) and grade 1+ rectal pain (*p* = 0.008) were significantly associated with developing late GI toxicity. The remainder of baseline and treatment factors demonstrated no association with grade 2+ late GI toxicity (6–24 months). In multivariable analysis, acute grade 2+ GI toxicity (OR 4.61, CI (2.11–10.06), *p* < 0.0001) remained statistically significantly associated with grade 2+ late GI toxicity (6–24 months). The AUC for the multivariable model’s ability to predict grade 2+ late GI (6–24 months) toxicity after CRT was 0.64 (95% CI, 0.57–0.72). Univariable analysis for predicting persistent late grade 2+ late GI events is summarised in Table 5. Due to low number of events multivariable analysis was not performed. Appendix A in the Appendix A summarises logistic regression analysis to predict late grade 2+ GI toxicity after 12 months.

### 3.3. Other Predictor Variables

The use of SBRT-CL rather than CyberKnife was significantly associated with increased odds of grade 2+ late GU toxicity after six months (OR 2.30, CI (1.34–3.94), *p* = 0.002). In multivariable analysis, SBRT-CL use was not significantly associated with persistent grade 2+ GU late toxicity and grade 2+ late GI toxicity. Multivariable logistic regression also demonstrated that fiducial use was associated with late grade 2+ GU toxicity after six months in the SBRT-CL arm (OR 2.44 (CI (1.35–4.40), *p* = 0.002). Similar results were seen when modelling for persistent late grade 2+ GU toxicity and grade 2+ GU toxicity after 12 months. No association between fiducial use and late-grade 2+ GI toxicity was seen. All those treated with CyberKnife radiotherapy had fiducials inserted. In a non-randomised assessment of the SBRT-CL (*n* = 242) population, there was greater proportion of worst CTCAE grade 2+ GU toxicity (after six months) in those with fiducials compared to those without (48% vs. 25%). In patients treated with SBRT-CL and fiducials (*n* = 130), 28% reported persistent grade 2+ GU late toxicity compared with 12% in those treated with SBRT-CL without fiducials (*n* = 111). However, in those treated with SBRT-CL and fiducials the median worst IPSS score in the late time period was 11 (IQR 6–18) vs. 12 (IQR 6–17) in those without fiducials.

### 3.4. Association between PROS and CROS

Appendix A in the Appendix A represent the relationship between maximum grade CTCAE toxicity and PROS (IPSS/EPIC-26 urinary/bowel bother) across acute and late timepoints. Appendix A (Appendix A) shows that as the maximum CTCAE grade increases from 0 to 1 and 2+, the median IPSS total score consistently rises throughout all time points. Appendix A (Appendix A) show that as maximum grade CTCAE toxicity increases, there is a consistent drop in median EPIC-26 urinary/bowel bother scores throughout the acute and late time. The weighted kappa tests agreement between patient-reported outcomes and CTCAE ranged from 0.25–0.35 (IPSS), 0.08–0.32 (EPIC-26 urinary bother) and 0.20–0.44 (EPIC-26 bowel bother), (Table 6).

## 4. Discussion

To our knowledge, this is one of the first studies assessing the association between acute and late toxicity in patients receiving SBRT and CRT for localised prostate cancer. We demonstrate a significant independent association between acute GI/GU and late GI/GU toxicity in patients treated with SBRT and CRT within the PACE-B study. In the SBRT and CRT populations, acute urinary toxicity and baseline urinary symptoms reported with either clinician or patient reported outcomes are associated with grade 2+ late GU toxicity endpoints. Acute and baseline grade 2+ GU events remained significantly associated with late grade 2+ GU toxicity in multivariable analysis in both treatment arms. Similarly, acute grade 2+ GI toxicity is significantly associated with late grade 2+ GI toxicity, following SBRT and CRT. In univariate analysis, baseline EPIC-26 bowel sub-domain scores were associated with CTCAE grade 2+ late GI toxicity following SBRT.

This association between acute and late toxicity persists despite multiple definitions of late toxicity, including a later timepoint from 12 months (to exclude the influence of acute residual events) and persistent late symptoms (to capture more severe late toxicity).

In univariable analysis, acute grade 2+ obstructive symptoms, irritative symptoms, dysuria/pain and urinary incontinence were predictive for late urinary toxicity following SBRT. In the CRT group, acute Grade 2+ obstructive symptoms and urinary incontinence were predictive for late GU toxicity. In univariable analysis, grade 1+ rectal pain, rectal bleeding, proctitis and grade 2+ diarrhoea were individually associated with grade 2+ late bowel toxicity following SBRT. In the CRT group, Grade 1+ rectal pain was associated with grade 2+ late bowel toxicity.

The area under ROC curve (AUC) for multivariable logistic regression modelling in these analyses were generally acceptable (AUC between 0.73–0.77) when modelling late urinary toxicity after CRT and SBRT (except for predicting for CTCAE grade 2+ GU late toxicity following CRT). However, the models’ performance was worse for predicting late bowel toxicity (AUC 0.64–0.66) (Appendix A). The difference in model performance between GU and GI toxicity could be due to greater GU toxicity events post-radiotherapy. Other explanation includes the absence of dosimetric data, and other patient factors not incorporated in the model, e.g., prior transurethral resection of the prostate, comorbidities and genetic factors.

Despite this correlation between acute and late toxicity, a significant proportion of patients with late grade 2+ toxicity did not experience grade 2+ acute toxicity. Therefore, patients can still develop late symptoms despite no or minimal acute toxicity potentially due to damage to late-reacting tissues or developing new urinary or bowel pathology subsequent to radiotherapy, triggering a CTCAE event. For example, late GI symptoms recorded on CTCAE measurements could be due the development of new pathologies such as small intestinal bacterial overgrowth, bile acid malabsorption, carbohydrate malabsorption etc. [23,24].

Acute and late toxicity are considered separate entities with different underlying pathogeneses. Acute toxicity often occurs within 1 to 2 weeks of starting radiotherapy and is usually reversible. It occurs generally due to loss of regenerative tissue stem cells, causing an imbalance between cell production and cell death in rapidly proliferating tissue such as the GI tract and skin. On the other hand, late toxicity manifests over six months to years after completing radiotherapy and is a consequence of fibrosis, atrophy and vascular injury, and is generally considered to be irreversible [25]. However, consequential late effects (CLE) can occur whereby acute radiation tissue injury can impair barriers against mechanical or chemical stress, causing a non-healing acute effect which develops into late effects [18]. This phenomenon could account for the association between acute and late toxicity. However, it is possible that the observed effect might be due to common characteristics which predispose to both acute and late toxicity.

Most studies demonstrating the association between acute and late toxicity in prostate cancer are based on patients treated with 3D conformal techniques and in patients treated with standard fractionation. [12] Heemsbergen et al. demonstrated in 553 patients treated in the 3D conformal Dutch dose escalation trial using multivariable analysis that acute mucous discharge (HR 1.8, *p* < 0.0001) and maximum acute proctitis (HR 1.8, *p* < 0.0001) were independently predictive for late grade 2+ toxicity [13]. Zelefsky et al. showed that in 1571 patients treated with 3D conformal radiotherapy and IMRT, grade 2 or worse CTCAE acute GI toxicity strongly predicted grade 2 or worse late GI toxicity (HR 6.95, *p* < 0.001). Similarly, acute grade 2+ GU toxicity strongly predicted late toxicity (HR 3.22, *p* < 0.001) [16]. Pinkawa et al. also showed that EPIC-bowel and urinary bother score changes in the acute period independently predicted late adverse urinary and bowel quality of life scores [17].

However, data supporting a correlation between acute and late toxicity is less conclusive after moderate hypofractionation. The CHHiP trial showed greater RTOG grade 2+ acute GI toxicity with the hypofractionated schedule (38% in both hypofractionated arms vs. 25% in the conventional arm). However, there was no difference in cumulative incidence of RTOG grade 2 or worse GI/GU events at five years [7]. Similarly, the PROFIT trial also reported higher proportions of cumulative acute grade 2+ GI effects (*p* = 0.003) in the hypofractionated arm. In the late period, grade 2+ late GI toxicity was better in the hypofractionated arm [8]. To our knowledge, multivariable analysis with acute toxicity as a predictor variable has not been performed in these datasets. Arcangeli et al. demonstrated that in 168 patients randomised to either conventional or moderately hypofractionated radiotherapy, there was a significant association between acute and late toxicity in the conventional arm. At the same time, there was no difference in the hypofractionated arm [26].

Other variables predictive of late grade 2+ GU toxicity include using the conventional linac for SBRT rather than CyberKnife. This has been discussed in more detail elsewhere [11]. There are several confounding factors with this comparison. For example, CyberKnife centres were large-volume academic centres and early SBRT adaptors with differences in baseline characteristics. Also, by chance there were more low risk patients and less patients on alpha-blockers at baseline in the CyberKnife arm. Further dosimetric analysis is planned to understand the interplay between late toxicity and treatment platform. In multivariable analysis, fiducial use was paradoxically associated with late GU toxicity in patients receiving SBRT, but not CRT. No association was seen with fiducial use and late GI toxicity in both arms. Further exploration showed that in the SBRT-CL population, CTCAE late GU toxicity was higher with fiducials than without, however there was no difference in worst IPSS scores between the two groups. The reasons for the correlation between toxicity rates and fiducial use with SBRT are not clear, however further analysis is planned to assess a centre effect. Confirmation of these differences seen between treatment platform and fiducial use can be performed in the PACE-C dataset.

The agreement between patient-reported outcomes (IPSS, EPIC-26 bladder and bowel bother) and worst grade 2+ CTCAE toxicity in all timepoints was fair. Rammant et al. assessed the agreement between EORTC QLQ-C30 and PR-25, and CTCAE and reported a similar mean kappa of 0.31 [27]. There are inherent differences between PROs and CROs [28]. For example, IPSS focuses on the frequency of urinary symptoms, while CTCAE grading depends on the effect symptoms have on activities of daily living and the degree of medical intervention required [29]. Patients may also score symptoms with greater severity than clinicians [30]. Also, the visits were compared (e.g., baseline, week 4, week 12 etc.) rather than actually dates for simplicity and to be as inclusive as possible. Some differences between CTCAE and patient reported outcomes could have due to the data not being collected at exactly the same time. As the weighted Kappa was low, it demonstrates the need to continue reporting patient and clinician-reported outcomes and possibly shift to patient-reported outcome measures as primary endpoints.

One strength of this study relates to the trial design of the PACE-B study. This is the largest phase 3 randomised control trial comparing SBRT and CRT for localised prostate cancer, providing a wealth of data for patients treated with both modalities. We demonstrate the association between acute and late GU/GI toxicities in the SBRT and CRT populations. We also use multiple definitions for late toxicity to exclude acute residual effects and ensure that more severe toxicity endpoints are tested. The incidence of grade 3 events was too low to include as a predictor and endpoint in logistic regression modelling. Therefore, persistent grade 2 or worse symptoms was used as marker for more severe toxicity. This work provides justification for using acute toxicity after SBRT as a predictor for late toxicity, potentially speeding up radiotherapy innovation.

One of the limitations of our work is that toxicity data was not complete for all patients, particularly patient-reported outcome measures (Appendix A). Another limitation is that the multivariable logistic regression models were not validated externally, and they lack dosimetric data. The association between acute and late effects was not adjusted for potential radiotherapy dose-volume interactions. In the Appendix A (Appendix A), there was no association between single dose-volume parameters for the rectum and bladder and acute and late GI/GU toxicity. More comprehensive dosimetric analyses are planned. Other potential predictor variables are unaccounted for in this model, e.g., previous trans-urethral resection of the prostate, patient comorbidities and genetic factors. Future work could entail validating the association between acute and late toxicity in other large prostate SBRT/CRT clinical trials including PACE-C and including dosimetric data and later time points.

We show that baseline symptoms are independently associated with increased odds of late toxicities. At baseline, many patients who were on alpha-blockers or anti-cholinergic were not scored as CTCAE Grade 2, thus subsequent reporting of the use of these medications will have triggered a Grade 2 event but actually reflects the status quo. Despite this, optimising symptoms pre-radiotherapy could mitigate late side effects, however further exploration is required. The increased odds of late toxicity after experiencing acute toxicity provides an opportunity to identify patients at higher risk of toxicity. Those reporting significant acute toxicity could be provided supportive care and early intervention to mitigate late side effects. However, it is important to remember that some patients still report late toxicity despite never experiencing significant acute side-effects.

## 5. Conclusions

Our study demonstrates an independent association between acute and late GU/GI toxicity in patients treated with SBRT and CRT for localised prostate cancer. Patients who experience grade 2+ acute GU/GI toxicity after prostate radiotherapy are at greater odds of developing late GU/GI toxicity. This can provide an opportunity for early intervention. The presence of grade 2 or worse acute toxicity should be considered an important variable for predicting late GI/GU toxicity after prostate radiotherapy.

## Figures and Tables

**Figure 1 cancers-15-01288-f001:**
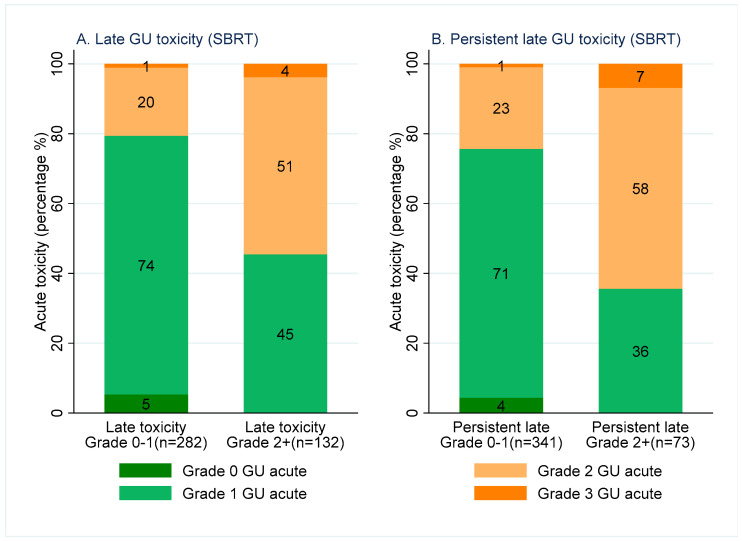
Stacked bar graph: maximum grade CTCAE acute GU toxicity experienced for patients with Grade 0–1 (**left** of each figure) and Grade 2+ (**right** of each figure) late GU toxicity following SBRT (**left**). Maximum grade CTCAE acute GU toxicity experienced by patients with grade 0–1 (**left** of each figure) and grade 2+ (**right** of each figure) persistent late GU toxicity following SBRT (**right**).

**Figure 2 cancers-15-01288-f002:**
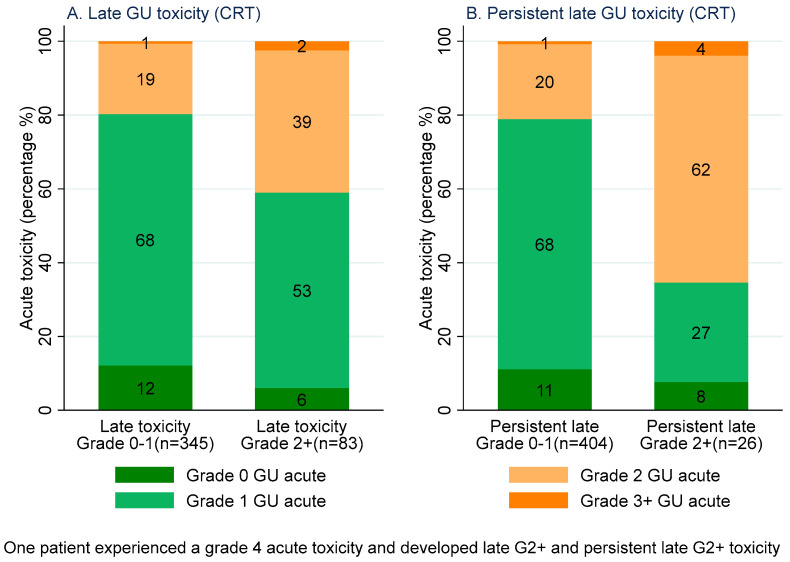
Stacked bar graph: maximum grade CTCAE acute GU toxicity experienced for patients with Grade 0–1 (**left** of each figure) and Grade 2+ (**right** of each figure) late GU toxicity following CRT (**left**). Maximum grade CTCAE acute GU toxicity experienced by patients with grade 0–1 (**left** of each figure) and grade 2+ (**right** of each figure) persistent late GU toxicity following CRT (**right**).

**Figure 3 cancers-15-01288-f003:**
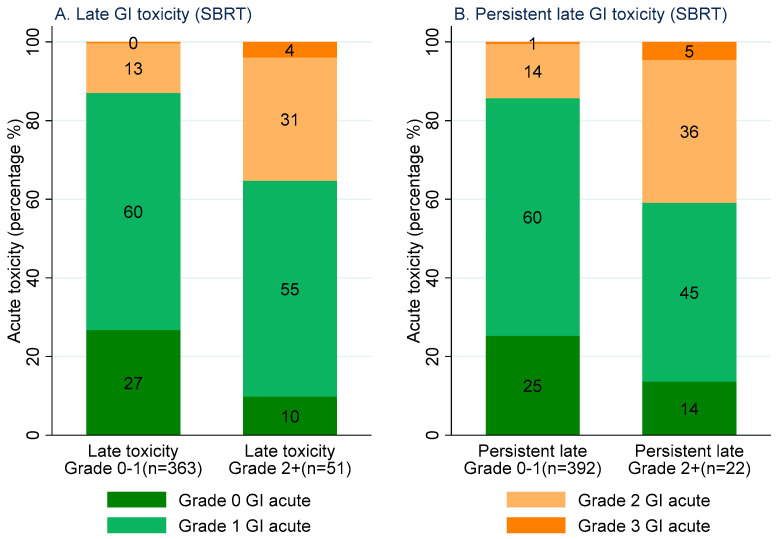
Stacked bar graph: maximum grade CTCAE acute GI toxicity experienced for patients with Grade 0–1 (**left** of each figure) and Grade 2+ (**right** of each figure) late GI toxicity following SBRT (**left**). Maximum grade CTCAE acute GI toxicity experienced by patients with grade 0–1 (**left** of each figure) and grade 2+ (**right** of each figure) persistent late GI toxicity following SBRT (**right**).

**Figure 4 cancers-15-01288-f004:**
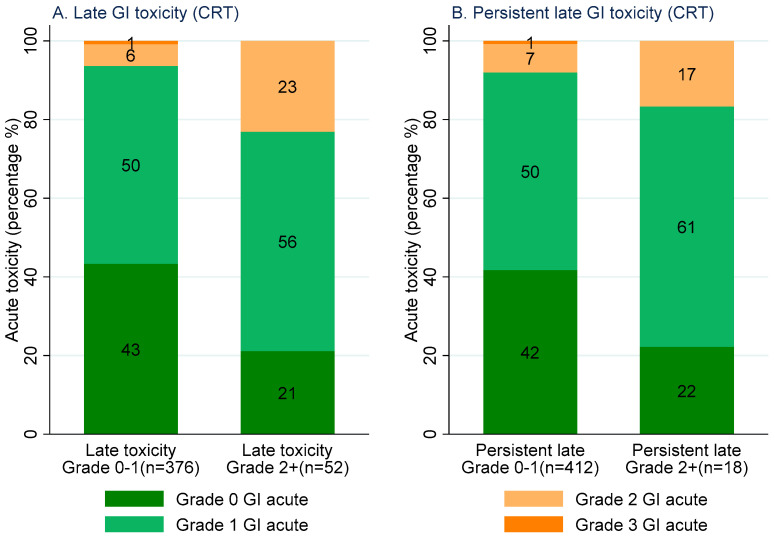
Stacked bar graph: maximum grade CTCAE acute GI toxicity experienced for patients with Grade 0–1 (**left** of each figure) and Grade 2+ (**right** of each figure) late GI toxicity following CRT (**left**). Maximum grade CTCAE acute GI toxicity experienced by patients with grade 0–1 (**left** of each figure) and grade 2+ (**right** of each figure) persistent late GI toxicity following CRT (**right**).

**Table 1 cancers-15-01288-t001:** Baseline demographics and clinical characteristics.

	Stereotactic Body Radiotherapy Group (n = 414)	Conventional Fractionated or Moderately Hypofractionated Radiotherapy (n = 428)
Median age in years	69 (65–73)	69 (65–73)
NCCN risk group		
low	35 (8%)	43 (10%)
intermediate	379 (92%)	885 (90%)
Performance status		
0	372 (90%)	379 (88.5%)
1	42 (10%)	47 (11%)
2	-	2 (0.5%)
Gleason Score		
3 + 3	61 (15%)	84 (20%)
3 + 4	353 (85%)	344 (80%)
Median pre-randomisation PSA	8 (5.5–11)	8 (6.3–10.7)
Prostate volume		
<40ml	189 (45%)	163 (38%)
40–80ml	197 (48%)	223 (52%)
80ml +	23 (6%)	27 (6%)
Missing	5	15
Pre-randomisation use of Alpha-blocker		
Yes	77 (19%)	84 (20%)
No	337 (81%)	344 (80%)
Pre-randomisation use of anti-cholinergic		
Yes	17 (4%)	15 (4%)
No	397 (96%)	413 (96%)
Fiducial marker use		
Yes	303 (73.2%)	244 (57%)
No	111 (26.8%)	184 (43%)
SBRT technique		
Conventional linac (CL)	242 (58.5%)	-
CyberKnife	169 (40.8%)	-
Other	3 (0.7%)	-

Data is presented in median (IQR), *n* (%). NCCN = National comprehensive cancer network. PSA = prostate-specific antigen. SBRT = Stereotactic body radiotherapy.

**Table 2 cancers-15-01288-t002:** Univariable and multivariable logistic regression model to identify covariates associated with CTCAE late genitourinary toxicity after SBRT.

Covariates	Level	CTCAE Grade 2+ GU Late Toxicity (6–24 Months)						CTCAE Persistent Grade 2+ GU Late Toxicity					
		No (n = 282)	Yes (n = 132)	Univariable		Multivariable (n = 404)		No (n = 341)	Yes (n = 73)	Univariable		Multivariable (n = 404)	
		N (%) or Median (IQR)	N (%) or Median (IQR)	OR (95% CI)	*p* value	OR (95% CI)	*p* value	N (%) or Median (IQR)	N (%) or Median (IQR)	OR (95% CI)	*p* value	OR (95% CI)	*p* value
Baseline grade 2+ GU symptoms	No Yes Missing	275 (98) 5 (2) 2	116 (88) 16 (12) 0	7.59 (2.72-21.19)	<0.0001	4.27 (1.41-13.04)	0.01	332 (98) 7 (2) 2	58 (81) 14 (19) 0	11.25 (4.36-29.1)	<0.0001	5.95 (2.1-16.9)	0.001
Baseline IPSS score	Median (n)	5 (3-10) (n=243)	8 (4-14) (n=114)	1.09(1.05-1.13)	<0.0001			5 (3-11) (n=295)	7.5 (4-15) (n=62)	1.07 (1.03-1.12)	0.002		
Baseline EPIC-26 overall urinary bother score	Median (n)	100 (75-100) (n=264)	75 (50-100) (n=119)	0.98 (0.97-0.99)	<0.0001			100 (75-100) (n=316)	75 (50-100) (n=67)	0.98 (0.97-0.99)	<0.0001		
Worst acute grade 2+ GU Toxicity	No Yes	224 (79) 58 (21)	60 (45) 72 (55)	4.63 (2.96-7.25)	<0.0001	3.70 (2.39-5.98)	<0.0001	258 (76) 83 (24)	26 (36) 47 (64)	5.62 (3.28-9.63)	<0.0001	4.26 (2.36-7.68)	<0.0001
Persistent acute grade 2+ GU toxicity	No Yes	261 (93) 21 (7)	91 (69) 41 (31)	5.60(3.14-9.98)	<0.0001			314 (92) 27 (8)	38 (52) 35 (48)	10.71 (5.85-19.60)	<0.0001		
Worst acute IPSS score	Median (n)	12 (8-18) (n=273)	17 (11-23) (n=173)	1.07(1.04-1.11)	<0.0001			13 (8-18) (n=332)	16.5 (11-24) (n=70)	1.06 (1.03-1.10)	<0.0001		
Worst acute EPIC-26 overall urinary bother score	Median (n)	75(50-100) (n=275)	50 (25-75) (n=126)	0.98 (0.97-0.99)	<0.0001			75(50-100) (n=331)	50 (25-75) (n=70)	1.77 (1.00-3.16)	0.05		
Worst acute urinary symptoms Grade 2+ Obstructive Grade 2+ irritative Grade 2+ pain/dysuria Grade 2+ Incontinence Grade 1+ haematuria	No Yes No Yes No Yes No Yes No Yes	270 (96) 12 (4) 242 (86) 40 (14) 264 (94) 18 (6) 280 (99) 2 (1) 256 (91) 26 (9)	107 (82) 24 (18) 79 (60) 53 (40) 106 (81) 25 (19) 120 (92) 11 (8) 117 (89) 14 (11)	5.04 (2.43-10.5) 4.06 (2.50-6.58) 3.46 (1.81-6.60) 12.8 (2.80-58.78) 1.18 (0.59-2.34)	<0.0001 <0.0001 <0.0001 0.001 0.639			325 (96) 15 (4) 282 (83) 59 (17) 313 (92) 27 (8) 337 (99) 3 (1) 307 (90) 33 (10)	52 (71) 21 (29) 39 (53) 34 (47) 57 (78) 16 (22) 63 (86) 10 (14) 66 (90) 7 (10)	8.75(4.24-18.06) 4.17 (2.43-7.14) 3.25 (1.65-6.24) 17.8 (4.77-66.3) 0.99 (0.42-2.33)	<0.0001 <0.0001 0.001 <0.0001 0.98		
Age	Median (n)	69 (65-73) (n=282)	70 (66-74) (n=132)	1.02 (0.99-1.06)	0.15	1.01 (0.97-1.05)	0.59	69 (65-73) (n=341)	69 (67-73) (n=73)	1.01 (0.98-1.05)	0.47		
Prostate Volume (cm^3^)	Median (n)	39 (30-54) (n=278)	44 (34-61) (n=131)	1.01 (1.00-1.03)	0.02	1.01 (0.99-1.02)	0.36	40 (31-55) (n=337)	47 (33-62) (n=72)	1.01 (1.00-1.02)	0.05	1.00 (0.99-1.02)	0.50
Risk group (intermediate)	low Int	23 (8) 259 (92)	12 (9) 120 (91)	0.89 (0.43-1.84)	0.75			29 (8) 312 (92)	6 (8) 67 (92)	1.04 (0.41-2.60)	0.08		
Baseline urinary medication (Yes)	No Yes	228 (81) 54 (19)	99 (75) 33 (25)	1.41 (0.86-2.30)	0.17			277 (81) 64 (19)	50 (69) 23 (31)	1.99 (1.13-3.49)	0.017	1.79 (0.94-3.40)	0.08
SBRT modality (CL)	CK CL Other/ missing	130 (46) 151 (54) 1	39 (30) 90 (68) 3	1.99 (1.28-3.09)	0.002	2.30 (1.34-3.94)	0.002	148 (44) 192 (56) 1	21 (29) 49 (67) 3	1.80 (1.03-3.13)	0.04	1.89 (0.98-3.68)	0.06
Fiducial use (Yes)	No Yes	83 (29) 199 (71)	28 (21) 104 (79)	1.55 (0.95-2.53)	0.08	2.44 (1.35-4.40)	0.002	98 (29) 243 (71)	13 (18) 60 (82)	1.86 (0.98-3.54)	0.06	2.41 (1.14-5.11)	0.02

OR = odds ratio; IQR = interquartile range; CL = conventional linac; GU = genitourinary; IPSS = international Prostate Symptom Score; EPIC = Expanded Prostate Cancer Index Composite; CK = CyberKnife; Int = Intermediate.

**Table 3 cancers-15-01288-t003:** Univariable and multivariable logistic regression model to identify covariates associated with late CTCAE genitourinary toxicity after CRT.

Covariates	Level	CTCAE Grade 2+ GU Late Toxicity (6–24 Months)						CTCAE Grade 2+ GU Persistent Late Toxicity					
		No (n = 345)	Yes (n = 84)	Univariable		Multivariable (n = 411)		No (n = 406)	Yes (n-27)	Univariable		Multivariable (n = 427)	
		N (%) or Median (IQR)	N (%) or Median (IQR)	OR (95% CI)	*p* value	OR (95% CI)	*p* value	N (%) or Median (IQR)	N (%) or Median (IQR)	OR (95% CI)	*p* value	OR (95% CI)	*p* value
Baseline grade 2+ GU toxicity	No Yes missing	335 (98) 7 (2) 3	72 (86) 12 (14) 0	7.98 (3.03-20.96)	<0.0001	6.73 (2.19-20.7)	0.001	392 (97) 11 (3) 3	19 (70) 8 (30) 0	15.0 (5.41-41.63)	<0.0001	7.78 (2.43-24.85)	0.001
Baseline IPSS score	Median (n)	6 (2-11) (n=299)	8 (5-13) (n=75)	1.05 (1.01-1.09)	0.009			6 (3-11) (n=351)	10 (6-12) (n=27)	1.04 (0.99-1.10)	0.093		
Baseline EPIC-26 overall urinary bother score	Median (n)	100 (75-100) (n=321)	75 (50-100) (n=78)	0.98 (0.97-0.99)	<0.0001			100 (75-100) (n=377)	75 (50-100) (n=26)	0.99 (0.97-0.99)	0.03		
Worst acute grade 2+ GU Toxicity	No Yes missing	277 (80) 68 (20) 0	49 (59) 34 (41) 1	2.83 (1.69-4.71)	<0.0001	2.26 (1.30-3.95)	0.004	319 (79) 85 (21) 2	9 (35) 17 (65) 1	7.09 (3.05-16.46)	<0.0001	4.84 (1.98-11.83)	0.001
Persistent grade 2+ acute GU toxicity	No Yes	328 (95) 17 (5)	60 (71) 24 (29)	7.72 (3.91-15.22)	<0.0001			381 (94) 25 (6)	11 (41) 16 (59)	22.17 (9.31-52.79)	<0.0001		
Worst acute IPSS score	Median (n)	11.5 (7-18) (n=336)	17 (12-21) (n=83)	1.08 (1.05-1.12)	<0.0001			12 (7-18) (n=394)	17 (14-22) (n=27)	1.09 (1.03-1.15)	0.001		
Worst acute EPIC-26 overall urinarybother	Median (n)	75 (50-100) (n=333)	50 (25-75) (n=81)	0.97 (0.97-0.98)	<0.0001			75 (50-100) (n=389)	50 (25-75) (n=27)	0.97 (0.96-0.99)	<0.0001		
Worst acute urinary symptoms Grade 2+ obstructive Grade 2+ irritative Grade 2+ pain/dysuria Grade 2+ incontinence Grade 1+ haematuria	No Yes No Yes No Yes No Yes No Yes	328 (95) 17 (5) 297 (86) 48 (14) 332 (96) 13 (4) 341 (99) 4 (1) 335 (97) 10 (3)	69 (83) 14 (17) 60 (72) 23 (28) 78 (94) 5 (6) 78 (94) 5 (6) 77 (93) 6 (7)	3.91 (1.84-8.32) 2.37 (1.34-4.19) 1.64 (0.57-4.73) 5.46 (1.34-20.82) 2.62 (0.92-7.40)	<0.0001 0.003 0.36 0.01 0.07			384 (95) 20 (5) 341 (84) 63 (16) 388 (96) 16 (4) 398 (99) 6 (1) 389 (96) 15 (4)	15 (58) 11 (42) 18 (69) 8 (31) 24 (92) 2 (8) 23 (88) 3 (12) 25 (96) 1 (4)	14.1 (5.73-34.58) 2.31 (1.00-5.77) 2.02 (0.44-9.30) 8.65 (2.03-36.82) 1.04 (0.13-8.17)	<0.0001 0.05 0.37 0.003 0.97		
Age	Median (n)	69 (65-74) (n=345)	68.5 (65-73) (n=84)	1.00 (0.96-1.04)	0.95	0.97 (0.93-1.02)	0.22	69 (65-73) (n=406)	69 (66-75) (n=27)	1.03 (0.97-1.10)	0.38		
Prostate Volume (cm^3^)	Median (n)	43 (32-55) (n=333)	47 (37-68) (n=81)	1.02 (1.01-1.03)	0.002	1.02 (1.00-1.03)	0.009	43 (33-57) (n=389)	44 (33-60) (n=27)	1.00 (0.98-1.02)	0.82		
Risk group (intermediate)	low Int	37 (11) 308 (89)	6 (7) 78 (93)	1.56 (0.63-3.83)	0.33	1.73 (0.63-4.76)	0.29	42 (10) 364 (90)	1 (4) 26 (96)	3 (0.40-22.66)	0.29		
Baseline urinary medication (Yes)	No Yes	275 (80) 70 (20)	61 (73) 23 (27)	1.48 (0.86-2.56)	0.16	1.20 (0.64-.2.23)	0.57	322 (79) 84 (21)	17 (63) 10 (37)	2.25 (1.00-5.10)	0.05		
Fiducial use (Yes)	No Yes	144 (42) 201 (58)	41 (49) 43 (51)	0.75 (0.47-1.21)	0.24			168 (41) 238 (59)	19 (70) 8 (30)	0.30 (0.13-0.69)	0.005	0.43 (0.17-1.06)	0.07

OR = odds ratio; IQR = interquartile range; GU = genitourinary; IPSS = international Prostate Symptom Score; EPIC = Expanded Prostate Cancer Index Composite; Int = intermediate.

**Table 4 cancers-15-01288-t004:** Univariable and multivariable logistic regression model to identify covariates associated with late CTCAE gastrointestinal toxicity after SBRT.

Covariates	Level	CTCAE Grade 2+ GI Late Toxicity (6–24 Months)						CTCAE Grade 2+ GI Persistent GI Late Toxicity			
		No (n = 357)	Yes (n = 51)	Univariable		Multivariable (n = 413)		No (n = 392)	Yes (n = 22)	Univariable	
		N (%) or Median (IQR)	N (%) or Median (IQR)	OR (95% CI)	*p* value	OR (95% CI)	*p* value	N (%) or Median (IQR)	N (%) or Median (IQR)	OR (95% CI)	*p* value
Baseline grade 2+ GI toxicity	No Yes Missing	357 (99) 4 (1) 2	51 (100) 0 0					386 (99) 4 (1) 2	22 (100) 0 0		
Baseline EPIC-26 bowel sub-domain score	Median (n)	100 (92-100) (n=318)	96 (88-100) (n=46)	0.96 (93-0.99)	0.004			100 (92-100) (n=344)	96 (86-100) (n=20)	0.95 (0.92-0.99)	0.007
Worst acute grade 2+ GI Toxicity	No Yes	316 (87) 47 (13)	33 (65) 18 (35)	3.67 (1.91-7.03)	<0.0001	3.68 (1.89-7.17)	<0.0001	336 (86) 56 (14)	13 (59) 9 (41)	4.15 (1.70-10.17)	0.002
Persistent grade 2+ acute GI toxicity	No Yes	353 (97) 10 (3)	47 (92) 4 (8)	3.00 (0.91-9.96)	0.07			381 (97) 11 (3)	19 (86) 3 (14)	5.47 (1.41-21.24)	0.014
Worst acute EPIC-26 bowel sub-domain score	Median (n)	88 (75-96) (n=351)	79 (63-94) (n=48)	0.99 (0.97-1.00)	0.04			88 (75-96) (n=378)	75 (63-88) (n=21)	0.98 (0.96-0.99)	0.01
Worst acute bowel symptoms Grade 1+ rectal bleeding Grade 2+ diarrhoea Grade 1+ rectal pain Grade 2+ proctitis	No Yes No Yes No Yes No Yes	277 (76) 85 (24) 342 (95) 20 (5) 277 (77) 85 (23) 346 (95) 17 (5)	32 (63) 19 (37) 43 (84) 8 (16) 24 (47) 27 (53) 45 (88) 6 (12)	1.93 (1.04-3.59) 3.18 (1.32-7.66) 3.67 (2.01-6.69) 2.71 (1.02-7.24)	0.04 0.01 <0.0001 0.05			297 (76) 94 (24) 367 (94) 24 (6) 290 (74) 101 (26) 373 (95) 19 (5)	12 (55) 10 (45) 18 (82) 4 (18) 11 (50) 11 (50) 18 (82) 4 (18)	2.63 (1.10-6.29) 3.40 (1.07-10.83) 2.87 (1.20-6.83) 4.36 (1.34-14.16)	0.03 0.04 0.02 0.014
Age	Median (n)	69 (65-73) (n=363)	68 (64-73) (n=51)	0.99 (0.95-1.04)	0.70			69 (65-73) (n=392)	71 (65-74) (n=22)	1.02 (0.96-1.10)	0.46
Prostate Volume (cm^3^)	Median (n)	40 (31-56) (n=359)	39.5 (33-57) (n=50)	1.00 (0.99-1.02)	0.53			40 (31-56) (n=388)	39 (32-58)	1.01 (0.99-1.03)	0.56
Risk group (intermediate)	low Int	28 (8) 335 (92)	7 (14) 44 (86)	0.53 (0.22-1.27)	0.15	0.45 (0.17-1.18)	0.19	33 (8) 359 (92)	2 (9) 20 (91)	0.92 (0.21-4.11)	0.91
SBRT modality (CL)	CK CL Other/ missing	155 (43) 206 (57) 2	14 (27) 35 (69) 2	1.88 (0.98-3.61)	0.06	1.67 (0.77-3.60)	0.19	167 (43) 223 (57) 2	2 (9) 18 (82) 2	6.74 (1.54-29.45)	0.01
Fiducial use (Yes)	No Yes	93 (26) 270 (74)	18 (35) 33 (65)	0.63 (0.34-1.17)	0.15	0.73 (0.35-1.54)	0.41	102 (26) 290 (74)	9 (41) 13 (59)	0.51 (0.21-1.22)	0.13

OR = odds ratio; IQR = interquartile range; CL = conventional linac; GI = gastrointestinal; EPIC = Expanded Prostate Cancer Index Composite; CK = Cyberknife; Int = Intermediate. Multivariable analysis for CTCAE grade 2+ persistent late GI toxicity was noted performed due to low number of events.

**Table 5 cancers-15-01288-t005:** Univariable and multivariable logistic regression model to identify covariates associated with late CTCAE gastrointestinal toxicity after CRT.

Covariates	Level	CTCAE Grade 2+ GI Late Toxicity (6–24 Months)						CTCAE Grade 2+ GI Persistent GI Late Toxicity			
		No (n=377)	Yes (n=52)	Univariable		Multivariable (n = 428)		No (n=415)	Yes (n=18)	Univariable	
		N (%) or Median (IQR)	N (%) or Median (IQR)	OR (95% CI)	*p* value	OR (95% CI)	*p* value	N (%) or Median (IQR)	N (%) or Median (IQR)	OR (95% CI)	*p* value
Baseline grade 2+ GI toxicity	No Yes Missing	373 (99) 2 (1) 2	49 (98) 1 (2) 2					308 (99) 3 (1) 4	18 (100) 0 (0) 0		
Baseline EPIC-26 bowel sub-domain score	Median (n)	100 (96-100) (n=339)	100 (92-100) (n=46)	0.99 (0.95-1.02)	0.55			100 (96-100) (n=373)	96 (92-100) (n=16)	0.98 (0.93-1.04)	0.55
Worst acute grade 2+ GI Toxicity	No Yes Missing	352 (94) 24 (6) 1	40 (77) 12 (23) 0	4.4 (2.04-9.47)	<0.0001	4.61 (2.11-10.06)	<0.0001	379 (92) 33 (8) 3	15 (83) 3 (17) 0	2.30 (0.63-8.34)	0.21
Persistent grade 2+ acute GI toxicity	No Yes	374 (99) 3 (1)	48 (92) 4 (8)	10.39 (2.26-47.82)	0.003			409 (99) 6 (1)	17 (94) 1 (6)	4.01 (0.46-35.19)	0.21
Worst acute EPIC-26 bowel sub-domain score	Median (n)	92 (75-100) (n=354)	88 (67-96) (n=51)	0.98 (0.97-1.00)	0.04			92 (75-100) (n=390)	87.5 (67-92) (n=17)	0.97 (0.95-1.00)	0.02
Worst acute bowel symptoms Grade 1+ rectal bleeding Grade 2+ diarrhoea Grade 1+ rectal pain Grade 2+ proctitis	No Yes No Yes No Yes No Yes	319 (85) 57 (15) 395 (100) 0 (0) 333 (89) 43 (11) 368 (98) 8 (2)	39 (75) 13 (25) 30 (83) 6 (17) 39 (75) 13 (25) 49 (94) 3 (6)	1.87 (0.94-3.71) 3.72 (0.66-20.83) 2.58 (1.28-5.22) 2.82 (0.72-10.97)	0.08 0.14 0.008 0.14			347 (84) 65 (16) 406 (99) 6 (1) 357 (87) 55 (13) 401 (97) 11 (3)	13 (72) 5 (28) 18 (100) 0 17 (94) 1 (6) 18 (100) 0	2.05 (0.71-5.96) 0.38 (0.05-2.93)	0.19 0.35
Age	Median (n)	69 (65-73) (n=377)	69 (66-73.5) (n=52)	1.01 (0.96-1.05)	0.80			69 (65-73) (n=415)	70 (67-75) (n=18)	1.04 (0.96-1.12)	0.37
Prostate Volume (cm^3^)	Median (n)	44 (33-58) (n=367)	43 (32-55) (n=47)	0.99 (0.98-1.01)	0.50			44 (33-58) (n=401)	38 (30-46) (n=15)	0.98 (0.95-1.02)	0.32
Risk group (intermediate)	low int	42 (11) 335 (89)	1 (2) 51 (98)	6.39 (0.86-47.48)	0.07	6.32 (0.83-48.2)	0.08	43 (10) 372(89)	0 18 (100)		
Fiducial use (Yes)	No Yes	158 (42) 219 (58)	27 (52) 25 (48)	0.67 (0.37-1.19)	0.17	0.74 (0.41-1.35)	0.33	181 (44) 234 (56)	6 (33) 12(67)	1.55 (0.57-4.20)	0.39

OR = odds ratio; IQR = interquartile range; GI = gastrointestinal; EPIC = Expanded Prostate Cancer Index Composite; Int = Intermediate. Multivariable analysis for CTCAE grade 2+ persistent late GI toxicity was noted performed due to low number of events.

**Table 6 cancers-15-01288-t006:** Weight kappa statistic demonstrating agreement between CTCAE and patient reported outcome measures (EPIC-26 and IPSS) is different timepoints.

Patient Reported Outcome Measures	*n* = 842 * (%)	Timepoint	Weighted Kappa (95% CI)	Time Difference (Days) between CTCAE and PRO Completion (Mean, SD)
EPIC-26 urinary bother	800 (95)	Baseline	0.28 (0.25–0.31)	9 (20)
	700 (83)	Week 4	0.20 (0.15–0.24)	2 (11)
	760 (90)	Week 12	0.29 (0.26–0.31)	4 (19)
	630 (75)	6 months	0.31 (0.30–0.32)	4 (15)
	744 (88)	12 months	0.44 (0.41–0.47)	4 (18)
	659 (78)	24 months	0.41 (0.34–0.45)	5 (18)
EPIC-26 bowel bother	766 (91)	Baseline	0.08 (0.00–0.17)	9 (20)
	675 (80)	Week 4	0.31 (0.22–0.35)	2 (11)
	723 (86)	Week 12	0.30 (0.21–0.32)	4 (19)
	596 (71)	6 months	0.26 (0.22–0.28)	4 (15)
	691 (82)	12 months	0.31 (0.24–0.35)	4 (18)
	621 (74)	24 months	0.32 (0.27–0.38)	5 (18)
IPSS total	747 (89)	Baseline	0.26 (0.20–0.31)	9 (20)
	661 (80)	Week 4	0.29 (0.25–0.34)	3 (11)
	728 (86)	Week 12	0.25 (0.24–0.28)	4 (19)
	705 (84)	6 months	0.29 (0.27–0.38)	4 (15)
	720 (86)	12 months	0.35 (0.33–0.39)	5 (20)
	593 (70)	24 months	0.35 (0.28–0.44)	5 (17)

* Represented as proportion of patients included in the acute/late association analysis. EPIC-26 = Expanded Prostate Cancer Index Composite-26; IPSS = International prostate symptom score.

## Data Availability

Formal requests for data sharing are considered in line with ICR-CTSU procedures with due regard given to funder and sponsor guidelines. Requests are via a standard proforma describing the nature of the proposed research and extent of data requirements. Data recipients are required to enter a formal data sharing agreement that describes the conditions for release and requirements for data transfer, storage, archiving, publication, and intellectual property. Requests are reviewed by the Trial Management Group (TMG) in terms of scientific merit and ethical considerations including patient consent. Data sharing is undertaken if proposed projects have a sound scientific or patient benefit rationale as agreed by the TMG and approved by the Independent Data Monitoring and Steering Committee as required. Restrictions relating to patient confidentiality and consent will be limited by aggregating and anonymising identifiable patient data. Additionally, all indirect identifiers that could lead to deductive disclosures will be removed in line with Cancer Research UK Data Sharing Guidelines.

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
