# Peer review of "The Association between Acute and Late Genitourinary and Gastrointestinal Toxicities: An Analysis of the PACE B Study"

_cancers, 2023, doi:10.3390/cancers15041288_

Round 1

Reviewer 1 Report

This is a post-hoc analysis of the PACE B study, a randomized trial of SBRT vs conventional fractionation for patients with low and intermediate risk localized prostate cancer. Preliminary results from this study have been published elsewhere, but this study focuses on the impact of acute toxicity (in addition to other factors) on the eventual development of clinically significant late toxicity.

Overall the authors are to be commended for an excellent analysis of the interplay between acute and late toxicity in patients undergoing SBRT (as well as more protracted fractionations) for prostate cancer. The authors are uniquely situated to be able to perform this analysis, which has not been well defined in the SBRT patient population.

Strengths:

Very large data set with consistent, prospective data collection

Weaknesses:

Post-hoc analysis

No DVH data or analysis

Late toxicity defined only until 2 years

Introduction:

- This is well written and adequately sums up the current state of the literature. I have no specific criticisms or complaints.

Methods:

- Why was the late period defined only up until 24 months? Clearly some patients can develop late radiation toxicity after 24 months and the trial is designed to follow patients for at least 5 years.

- Was there substantial variation in the visit data assigned by sites compared to the actual date of visit or is this information unknown?

- Were margins different depending on the machine used (CK vs. Linac) for SBRT?

Results:

-       Similar to my question in the methods, what about toxicity occurring after 2 years? Clearly late GI/GU toxicity can occur >24 months out from completion of treatment.

-       What was the median follow-up for patients in this study. You state that patients only had to have at least one CTCAE grading in the acute and late periods. This suggests some patients may have only had a late toxicity evaluation at 6 months which seems inadequate for this analysis.

-       The labeling of the stacked bar graphs could be made more clear, particularly that the Y-axis reflects the percentage with acute toxicity

-       One thing I wonder about is the impact on late urinary symptom flare on this analysis, which has been previously reported in a substantial number of patients treated with SBRT. While it is bothersome to patients, those symptoms do resolve with conservative measures. What was the overall incidence of G2+ toxicity at the end of the follow-up period (per the recently published study from the same investigators, I would guess it is significantly lower than even what is reported here as persistent toxicity)? I suppose my question is how clinically significant these endpoints are if they resolve with conservative measures?

-       Do you have data regarding the need for invasive intervention (CTCAE G2 urinary toxicity can include medical management (urinary frequencygrading ) or more invasive measures (urinary retention or urinary tract obstruction grading

Discussion

-       I am not sure any of the examples provided in lines 525-527 to explain late GI symptoms are relevant to patients undergoing prostate radiation. I would rewrite with examples that are actually something that could be attributed to prostate radiation.

-       I agree that formal dose-volume or NTCP analysis of this dataset would be a great future analysis

-       I question the final concluding paragraph where you state that more intensive monitoring of these patients is warranted given the endpoints of the study. The endpoints studied, as best I can tell, are relatively easily managed when the arise in clinical practice, and if the prior PACE B publications are accurate, largely resolve by 24 months. What closer monitoring would you suggest for these patients?

Author Response

Comments and Suggestions for Authors

This is a post-hoc analysis of the PACE B study, a randomized trial of SBRT vs conventional fractionation for patients with low and intermediate risk localized prostate cancer. Preliminary results from this study have been published elsewhere, but this study focuses on the impact of acute toxicity (in addition to other factors) on the eventual development of clinically significant late toxicity.

Overall the authors are to be commended for an excellent analysis of the interplay between acute and late toxicity in patients undergoing SBRT (as well as more protracted fractionations) for prostate cancer. The authors are uniquely situated to be able to perform this analysis, which has not been well defined in the SBRT patient population.

Strengths:

Very large data set with consistent, prospective data collection

Weaknesses:

Post-hoc analysis

No DVH data or analysis

Late toxicity defined only until 2 years

Introduction:

- This is well written and adequately sums up the current state of the literature. I have no specific criticisms or complaints.

Methods:

- Why was the late period defined only up until 24 months? Clearly some patients can develop late radiation toxicity after 24 months and the trial is designed to follow patients for at least 5 years.

We thank you the reviewer for their comments and for reviewing the paper. Currently toxicity data beyond 2 years is unavailable. Other studies (e.g. the CHHiP trial) have shown that rates of new late toxicity events start to plateau after 2 years, so we feel this is a reasonable length of follow up to make preliminary conclusions.

- Was there substantial variation in the visit data assigned by sites compared to the actual date of visit or is this information unknown?

Thank you for raising this question. It was a pragmatic decision to use the scheduled visits as opposed to the actual date. The were some differences between the scheduled visit and the actual visit. However, the mean differences was 0.5 weeks (95% CI 0.4 to 0.5) for the acute period and 1 week (95% CI 0.9 to 1.1) for the late period, with the actual visits generally later than the schedule visits. 95% of the assessments fell within the schedule visit window assuming the window was half-way between each visit. As we are summing the data over the acute and late periods most importantly is that the visit dates fall within the relevant acute and late periods. For our data of the less than 1% fall outside of the relevant time period windows

- Were margins different depending on the machine used (CK vs. Linac) for SBRT?

The differences between SBRT and linac will be explored in another study. There was no difference in the median margins used between CyberKnife and Linac, except posteriorly. The median posterior margin for linac based SBRT was 3.5mm vs 3mm for CK.

Results:

-       Similar to my question in the methods, what about toxicity occurring after 2 years? Clearly late GI/GU toxicity can occur >24 months out from completion of treatment.

We only have 2-year follow-up data available, and the 5-year toxicity data is pending.

-       What was the median follow-up for patients in this study. You state that patients only had to have at least one CTCAE grading in the acute and late periods. This suggests some patients may have only had a late toxicity evaluation at 6 months which seems inadequate for this analysis.

-The median follow-up was 24.1 months (IQR 23.6-24.7). Table 4 in the supplementary section shows the patient level CTCAE completion rates. For late toxicity (from 6-24 months) we have at least 5 CTCAE late measurements (e.g. 6 months, 12 months, 18 months etc) in 92% of patients.

-       The labeling of the stacked bar graphs could be made more clear, particularly that the Y-axis reflects the percentage with acute toxicity

Thank you, I will update the graph.

-       One thing I wonder about is the impact on late urinary symptom flare on this analysis, which has been previously reported in a substantial number of patients treated with SBRT. While it is bothersome to patients, those symptoms do resolve with conservative measures. What was the overall incidence of G2+ toxicity at the end of the follow-up period (per the recently published study from the same investigators, I would guess it is significantly lower than even what is reported here as persistent toxicity)? I suppose my question is how clinically significant these endpoints are if they resolve with conservative measures?

This is an important point; we chose persistent late symptoms (2 or more grade 2+ events) as we felt that captures more clinically significant toxicity, even if it occurs within two years. This definition would potentially exclude those with a brief flare of symptoms occurring at only one follow up timepoint. We also repeated the analysis with a later endpoint (12-24 months) to exclude consequential toxicity which started as acute toxicity but persisted. Data capturing significant persistent late toxicity is unavailable currently but will be available in the coming years.

-       Do you have data regarding the need for invasive intervention (CTCAE G2 urinary toxicity can include medical management (urinary frequencygrading ) or more invasive measures (urinary retention or urinary tract obstruction grading

Urinary tract obstruction grading is not included. Whether grade 2+ urinary retention is due to medication started or insertion of a urinary catheter is unfortunately not directly recorded but we will examine whether we can infer this from the RTOG data (where catheter placement is a Grade 4 toxicity). In the supplementary section of the PACE-B acute toxicity paper in Lancet Oncology (Supplementary Table 12) it shows that 1 patient in the CRT arm and 2 patients in the SBRT arm were recorded as Grade 4 RTOG GU toxicity during the acute toxicity period). These patients were likely due to catheter insertion.

Discussion

-       I am not sure any of the examples provided in lines 525-527 to explain late GI symptoms are relevant to patients undergoing prostate radiation. I would rewrite with examples that are actually something that could be attributed to prostate radiation.

Thank you for your helpful comment. I have changed this, to state that late GI symptoms recorded on CTCAE measurements could be due to the development of new diseases such as small intestinal bacterial overgrowth, bile acid malabsorption etc.

-       I agree that formal dose-volume or NTCP analysis of this dataset would be a great future analysis

This analysis is planned and we hope to report data from this over the next year.

-       I question the final concluding paragraph where you state that more intensive monitoring of these patients is warranted given the endpoints of the study. The endpoints studied, as best I can tell, are relatively easily managed when the arise in clinical practice, and if the prior PACE B publications are accurate, largely resolve by 24 months. What closer monitoring would you suggest for these patients?

Thank you, on reflection I agree that this comment is unwarranted. I have removed closer monitoring from the conclusion.

Reviewer 2 Report

my suggestion for revisions are listed as follows:

-period considered for acute toxicity is defined up to 12 weeks and for late toxicicity from 6 months. What about the 3 to 6 months interval? Authors may better clarify this apect. 

-I think that differences in toxicity between techniques require a better explanation. Despite some interpretation has been made in previous pubblication, an interpretation at least of the trend toward correlation between robotic treatment and GI toxicity may be deeply analyzed. 

Author Response

-period considered for acute toxicity is defined up to 12 weeks and for late toxicicity from 6 months. What about the 3 to 6 months interval? Authors may better clarify this apect. 

We thank the reviewer for their comments. Each assessment collects events since the last assessment and therefore the 3-6 months is included in the 6 month visit, just as the 6-9 months is collected in the 9 month visit. There is of course some variation in the timing of the visits. Some additional wording has been added to clarify this and some results have been provide regarding the timeliness of the visits.

-I think that differences in toxicity between techniques require a better explanation. Despite some interpretation has been made in previous pubblication, an interpretation at least of the trend toward correlation between robotic treatment and GI toxicity may be deeply analyzed. 

This is an important question and we require further dosimetric analysis to clarify whether this relates to some aspect of the treatment machine (dosimetry, image guidance, tracking) or whether it reflects the confounding variables associated with Cyberknife centres vs non-Cyberknife centres. This an area of ongoing work, and it will be further explored in subsequent publications once we have further information. We have clarified this in the discussion section.

Reviewer 3 Report

The authors present a posthoc analysis of a prospective randomized clinical trial in 842 low and intermediate risk prostate cancer patients.

They analysed the association between early GI/GU toxicity and late GI/GU toxicity after ultrahypofractionated (SBRT) and conventional fractionated radiotherapy (CRT), while considering patient, tumour and treatment factors.

In result, they describe a significant independent association between acute GI/GU and late GI/GU toxicity. Acute and baseline grade 2+ GU events remained significantly associated with late grade 2+ GU toxicity in multivariable analysis. Similar they found that acute grade 2+ GI toxicity is significantly associated with late grade 2+ GI toxicity.

-          The authors could clarify, how they handled baseline toxicity (tox): A patient having G2 GU tox at baseline and G2 tox during follow up has no deterioration of symptoms: is this considered as no additional tox at follow up (G0), or did the authors not correct for baseline tox (G2 tox at follow up)? In the latter case it´s not surprising that baseline tox (and early tox, that did not improve from baseline) correlates with late tox.

-          The CTCAE 4.03 term “urinary retention, Grade 2” is defined as “…catheter placement indicated; medication indicated”: 20% of patients had alpha-blockers before randomization: were these counted as G2 GU tox at baseline?

-          Line 549: spelling mistake: “worse”?

the submission is well written,, the statistical analysis is sound. The results are of relevance for the community.

Author Response

The authors present a posthoc analysis of a prospective randomized clinical trial in 842 low and intermediate risk prostate cancer patients.

They analysed the association between early GI/GU toxicity and late GI/GU toxicity after ultrahypofractionated (SBRT) and conventional fractionated radiotherapy (CRT), while considering patient, tumour and treatment factors.

In result, they describe a significant independent association between acute GI/GU and late GI/GU toxicity. Acute and baseline grade 2+ GU events remained significantly associated with late grade 2+ GU toxicity in multivariable analysis. Similar they found that acute grade 2+ GI toxicity is significantly associated with late grade 2+ GI toxicity.

-          The authors could clarify, how they handled baseline toxicity (tox): A patient having G2 GU tox at baseline and G2 tox during follow up has no deterioration of symptoms: is this considered as no additional tox at follow up (G0), or did the authors not correct for baseline tox (G2 tox at follow up)? In the latter case it´s not surprising that baseline tox (and early tox, that did not improve from baseline) correlates with late tox.

We thank the reviewer for their helpful comments which we have carefully considered. We feel we have described this by explaining the multivariate analysis that we have undertaken. We have chosen not to include analysis on no deterioration in symptoms in this paper as we want to assess factors that can predict having grade 2+ toxicity during the late period. While having G2+ toxicity at baseline is likely to mean the patient will have this during the late toxicity this was not the case for all patients. And in the MVA both baseline and acute g2+ events were included in the lasso variable selection process and therefore this took into consideration the baseline data and whether the acute G2+ toxicity was still associated with the late G2+ toxicity.

-          The CTCAE 4.03 term “urinary retention, Grade 2” is defined as “…catheter placement indicated; medication indicated”: 20% of patients had alpha-blockers before randomization: were these counted as G2 GU tox at baseline?

We thank the reviewer for highlighting a very important point. Of the patients who had urinary medication (alpha or anticholinergic) at baseline (n=182), only 10% (n=19) were recorded as CTCAE grade 2+ at baseline. We have now included this important point in the results and discussion.

-          Line 549: spelling mistake: “worse”?

Thank you, this has now been changed

the submission is well written,, the statistical analysis is sound. The results are of relevance for the community.